# Concomitant Activation of OSM and LIF Receptor by a Dual-Specific hlOSM Variant Confers Cardioprotection after Myocardial Infarction in Mice

**DOI:** 10.3390/ijms23010353

**Published:** 2021-12-29

**Authors:** Holger Lörchner, Juan M. Adrian-Segarra, Christian Waechter, Roxanne Wagner, Maria Elisa Góes, Nathalie Brachmann, Krishnamoorthy Sreenivasan, Astrid Wietelmann, Stefan Günther, Nicolas Doll, Thomas Braun, Jochen Pöling

**Affiliations:** 1Department of Cardiac Development and Remodeling, Max Planck Institute for Heart and Lung Research, 61231 Bad Nauheim, Germany; Holger.Loerchner@mpi-bn.mpg.de (H.L.); asejuan@gmail.com (J.M.A.-S.); Christian.Waechter@mpi-bn.mpg.de (C.W.); Roxanne.Wagner@mpi-bn.mpg.de (R.W.); Maria.Goes@mpi-bn.mpg.de (M.E.G.); Nathalie.Brachmann@mpi-bn.mpg.de (N.B.); Krishna-Moorthy.Sreenivasan@mpi-bn.mpg.de (K.S.); Stefan.Guenther@mpi-bn.mpg.de (S.G.); 2German Centre for Cardiovascular Research (DZHK), Partner Site Rhein-Main, Frankfurt am Main, 60323 Frankfurt, Germany; 3Department of Cardiology, University Hospital Marburg, Baldingerstraße, 35043 Marburg, Germany; 4MRI and µ-CT Service Group, Max Planck Institute for Heart and Lung Research, 61231 Bad Nauheim, Germany; Astrid.Wietelmann@mpi-bn.mpg.de; 5Department of Cardiac Surgery, Schüchtermann-Clinic, Ulmenallee 5-11, 49214 Bad Rothenfelde, Germany; Ndoll@schuechtermann-klinik.de

**Keywords:** myocardial infarction, cardiac remodeling, cardioprotection, cytokine signaling

## Abstract

Oncostatin M (OSM) and leukemia inhibitory factor (LIF) signaling protects the heart after myocardial infarction (MI). In mice, oncostatin M receptor (OSMR) and leukemia inhibitory factor receptor (LIFR) are selectively activated by the respective cognate ligands while OSM activates both the OSMR and LIFR in humans, which prevents efficient translation of mouse data into potential clinical applications. We used an engineered human-like OSM (hlOSM) protein, capable to signal via both OSMR and LIFR, to evaluate beneficial effects on cardiomyocytes and hearts after MI in comparison to selective stimulation of either LIFR or OSMR. Cell viability assays, transcriptome and immunoblot analysis revealed increased survival of hypoxic cardiomyocytes by mLIF, mOSM and hlOSM stimulation, associated with increased activation of STAT3. Kinetic expression profiling of infarcted hearts further specified a transient increase of OSM and LIF during the early inflammatory phase of cardiac remodeling. A post-infarction delivery of hlOSM but not mOSM or mLIF within this time period combined with cardiac magnetic resonance imaging-based strain analysis uncovered a global cardioprotective effect on infarcted hearts. Our data conclusively suggest that a simultaneous and rapid activation of OSMR and LIFR after MI offers a therapeutic opportunity to preserve functional and structural integrity of the infarcted heart.

## 1. Introduction

The onset of myocardial infarction (MI) features a rapid but transient inflammatory response at sites of injury. Infiltrating immune cells and cardiac resident cells subsequently release a plethora of cytokines, whose task is to initiate, adjust and terminate cardiac repair events [1]. Insufficient and excessive expression of inflammatory cytokines are associated with impaired myocardial healing and progression to heart failure, respectively [2,3]. Cytokines, therefore, have gained interest as potential targets for the treatment of MI. Certainly, inhibition strategies of individual cytokines as the prevailing therapeutic approach in clinical trials mostly produced disappointing and inconsistent results [4].

Recent studies by our group and others demonstrated a cardioprotective role of the IL–6 type cytokines oncostatin M (OSM) and leukemia inhibitory factor (LIF) in rodent models of MI [5,6,7,8,9,10]. Pre-infarction delivery of recombinant cytokines and plasmid DNA were associated with an improved cardiac function and restricted infarct expansion in mice [5,7,8,9]. Such cytokine delivery strategies might complement conventional therapies to support endogenous repair events but more detailed studies about temporo-spatial activity of different cytokine circuits during myocardial healing is needed. Specifically, it is necessary to define the time frame for putative therapeutic interventions and to minimize any potential collateral damage arising from inadequate activation.

OSM and LIF share a common evolutionary origin and structure [11,12,13]. Both cytokines communicate and confer instructions to target cells via binding to their cognate receptors oncostatin M receptor (OSMR) and leukemia inhibitory factor receptor (LIFR) [12]. Human OSM is additionally able to bind and signal via LIFR, while mouse OSM exclusively binds to OSMR [14]. The formation of oligomeric receptor complexes finally initiate intracellular signaling events through phosphorylation of components of the Janus kinase/signal transducer and activator of transcription (Jak/STAT) pathway, the mitogen-activated protein kinases (MAPKs) pathways and the phosphoinositide 3-kinase/protein kinase B (PI3K/Akt) pathway [12]. Intracellular crosstalk with other transcription factors such as nuclear factor-κB (NF-κB) and c–Myc has been reported as well [15,16].

The aforementioned species-specific receptor binding profile of OSM limits the usefulness of the murine system to mimic human conditions, thereby preventing direct translation of preclinical data obtained in the mouse into potential clinical trials and applications. Previous work by our group showed that changes in the amino acid composition of the so-called AB loop in OSM determines this species-specific binding diversity [17]. By replacing the amino acid sequence of the murine AB loop by the corresponding human protein, we engineered a human-like OSM (hlOSM) mutant protein that is capable to signal both via the OSMR and LIFR in murine cells [17]. In this study, we compared the effects of selective LIFR– and OSMR– signaling to concomitant activation of LIFR and OSMR in normoxic and hypoxic murine cardiomyocytes. Based on a detailed profiling of the temporospatial expression pattern of OSM and LIF during myocardial healing in mice, we additionally demonstrate for the first time that a short-term, post-infarction delivery of recombinant hlOSM was superior compared to administration of mLIF or mOSM alone to improve cardiac architecture and contractility after MI.

## 2. Results

### 2.1. Transcriptome and Gene Set Enrichment Analysis Characterizes STAT3, STAT5 and c–Myc as Major Common Signaling Molecules Downstream of OSMR and LIFR Activation in Cardiomyocytes

Human OSM is able to bind and signal via the OSMR and LIFR, while mouse OSM exclusively binds to OSMR [12,14] (Figure 1A). Changes in the amino acid composition of the AB loop were recently identified to determine this species-specific binding diversity [17]. By replacing the amino acid sequence of the murine AB loop by the corresponding human protein, we were able to create a human-like OSM (hlOSM) mutant protein that is capable to signal via the OSMR and LIFR in murine cells as reported previously [17] (Figure 1A,B). The possibility to induce exclusive LIFR–, OSMR– and dual LIFR/OSMR–mediated signaling in murine cardiomyocytes prompted us to characterize downstream effects of murine LIF (mLIF), murine OSM (mOSM) and the hlOSM mutant via transcriptome analysis (Figure 2A). Accordingly, RNA was isolated from murine cardiomyocytes stimulated with mLIF, mOSM and hlOSM for 24 h (Figure 2A). Addition of equivalent volumes of sterile PBS served as controls.

Principal component analysis visualized a separate clustering of differentially expressed genes (DEG) in PBS– and mLIF–treated cardiomyocytes, while the set of DEG in mOSM– and hlOSM–treated cardiomyocytes partially overlapped (Figure 2B). Moreover, the number of down- and up-regulated genes in mOSM– and hlOSM-treated cardiomyocytes versus control exceeded approximately ten-fold the number of genes being regulated by mLIF (Figure 2C). Along these lines, the majority of up- and down-regulated DEG were shared by mOSM and hlOSM, whereas the number of DEG shared by mLIF with either mOSM or hlOSM was very limited (Figure 2D,E). Overall, we found 50 down- and 36 up-regulated genes by all three cytokines when compared to PBS-treated cardiomyocytes (Figure 2D,E, Appendix A). The highest number of genes being exclusively up- and down-regulated by a singular cytokine was referred to hlOSM (Figure 2D,E). 

We next performed gene set enrichment analysis (GSEA) of DEG obtained from pairwise comparisons of all cytokine-treated versus control samples to specify downstream signaling cascades and molecular functions that are induced and affected by mLIF, mOSM and hlOSM, respectively. Based on an unsupervised, hierarchical clustering of DEG ranked by a normalized enrichment score, particularly gene clusters related to STAT3- and STAT5-mediated signaling cascades, whose activation has been reported to play a pivotal cardioprotective role, were increased in mLIF–, mOSM– and hlOSM–treated cardiomyocytes (Figure 2F) [18,19,20,21,22,23]. Target genes of the transcription factor c–Myc and inflammatory-associated genes (e.g., Cxcl9, Il6, Timp1) were also found to be positively enriched in cardiomyocytes upon stimulation with mLIF, mOSM and hlOSM (Figure 2F).

Vice versa, GSEA identified a reduced number of myogenic, sarcomeric genes, which is most likely reflecting a phenotypic conversion of cardiomyocytes undergoing dedifferentiation as reported previously by our group (Figure 2F) [5,24].

A restricted number of gene clusters were individually deregulated upon mLIF, mOSM or hlOSM stimulation and were especially related to inflammatory and stress-related processes such as TNF-α signaling via NFκB, INF-α response and UV-induced DNA response, respectively (Figure 2F).

### 2.2. OSMR– and LIFR–Mediated Activation of STAT3 but Not of STAT5 and c–Myc Is Linked to Increased Survival of Cultured Cardiomyocytes under Hypoxic Conditions

Our functional enrichment analysis of murine cardiomyocytes suggested that mLIF, mOSM and hlOSM commonly activate signaling cascades, which in turn are supposed to protect the heart under ischemic conditions [25]. We, therefore, aimed to substantiate this hypothesis by monitoring the profile of activated, i.e., phosphorylated (p–)STAT3 (Tyr705), p–STAT5 (Tyr694) and p–c–Myc (Ser62) in PBS– versus mLIF–, mOSM– and hlOSM-treated cardiomyocytes cultured under normoxic (21% O_2_) and hypoxic (1% O_2_) conditions. Accordingly, addition of mLIF, mOSM and hlOSM to both normoxic and hypoxic cardiomyocytes induced an increase of p-STAT3 within 24 h (Figure 3A,B). The human-like OSM mutant protein exerted the most potent effect on STAT3 activation, whereas mLIF was least pronounced among all cytokines tested (Figure 3A,B). The activation profile of STAT5 and c–Myc was comparable to STAT3, identifying hlOSM as the strongest inducer, followed by mOSM and mLIF (Figure 3C–F). A statistically significant increase of p–STAT5 and p–c–Myc, however, was only detectable under normoxic but not hypoxic conditions (Figure 3C–F).

Next, we directly tested the capability of mLIF, mOSM and hlOSM to protect cardiomyocytes from hypoxia-mediated cell death in vitro. Isolated murine cardiomyocytes were treated with PBS, mLIF, mOSM and hlOSM and exposed to hypoxic conditions. After 24 h, the viability dye Calcein-AM and nuclei dye Hoechst 33342 were added to quantify the percentage of viable cardiomyocytes (Figure 3G). PBS-treated cardiomyocytes cultured under normoxic conditions were used as a reference. Thus, we could identify a significant decrease of viable cardiomyocytes in PBS-treated cardiomyocytes maintained under hypoxic versus normoxic conditions (Figure 3G,H). The addition of mLIF, mOSM and hlOSM to cell culture medium efficiently prevented this substantial loss of Calcein^pos^ cardiomyocytes, whereas the human-like OSM mutant protein was most effective in suppressing hypoxia-mediated cell death (Figure 3G,H). We complementarily measured the release of lactate dehydrogenase by hypoxic cardiomyocytes, which is proportional to the amount of damaged or injured cells. Here, mLIF, mOSM and hlOSM were all shown to significantly reduce the amount of lactate dehydrogenase in a comparable manner (Figure 3I). Our in vitro studies generally underline a direct cardioprotective effect of mOSM, mLIF and hlOSM on cardiomyocytes. The maximum increase of activated STAT3 and suppression of cell death under hypoxic conditions via the human-like OSM mutant further imply additive effects via simultaneous activation of OSMR and LIFR.

### 2.3. Kinetic Expression Pattern of OSM, LIF and Their Corresponding Receptors in Cardiac Tissue after the Onset of Myocardial Infarction in Mice

A comparative assessment of therapeutic-oriented, interventional delivery strategies of OSM and LIF to support endogenous repair events has not been conducted yet. In order to delineate the time window for intervention, we profiled the temporal expression pattern of OSM, LIF and their corresponding receptors during myocardial healing in mice. A permanent surgical ligation of the left anterior descending coronary artery in adult male C57/Bl6 mice was employed to mimic myocardial infarction (MI) in humans. Infarcted hearts were collected at Days 1, 2, 4, 7 and 14 post-MI encompassing the early inflammatory phase and the consecutive reparative phase of cardiac repair in mice [26] (Figure 4A). We additionally fractionated hearts into a non-infarcted remote zone (RZ) and infarction zone (IZ) to identify potential spatial differences by means of immunoblot analysis (Figure 4A). Within the first 2 days post-MI, we observed comparable expression levels of the OSMR in RZ and IZ (Figure 4B,C). From Day 4 post-MI on, we identified a marked and significant increase of the OSMR at sites of injury (Figure 4B,C). These findings contrasted with the temporospatial expression profile of the LIFR, displaying a moderate but gradual decline within the IZ during our period of observation (Figure 4B,D). With regard to both ligands, we identified an equivalent and sustained increase of OSM and LIF within the IZ during the early inflammatory phase (Figure 4B,E,F). Intriguingly, we also observed a comparable expression profile of OSM and LIF in blood of infarcted mice, wherein both cytokines continuously increased over time (Appendix A). 

We further studied the spatial expression profile of OSMR and LIFR in infarcted hearts of mice by immunohistochemical analysis. The overall expression pattern of both receptors was characterized by an intense staining of OSMR and LIFR within the infarcted region, which might be referred to the presence of fibroblasts as described previously (Figure 4G,H) [27]. Moreover, we found OSMR^pos^ and LIFR^pos^ cardiomyocytes allocated in close proximity to the infarcted area and within the remote zone, whereas the LIFR seems to be more evenly distributed within the myocardium (Figure 4G,H). Our kinetic expression profiling conclusively implies an increased activation of OSM- and LIF-mediated signaling in ischemic hearts during the early inflammatory phase. A comparable spatial expression profile of OSMR and LIFR expressing cardiomyocytes additionally argue for overlapping or additive ligand-receptor interactions. 

### 2.4. Post–Infarction Administration of mLIF, mOSM and hlOSM Modulates Activation of STAT3, STAT5 and c–Myc at Distinct Sites within the Myocardium

The protective effect of mLIF, mOSM and hlOSM on ischemically challenged cardiomyocytes, along with the endogenous transient increase of OSM and LIF in ischemic hearts during the early inflammatory phase conclusively led us to study the effects of a temporally restricted, systemic administration of mLIF, mOSM and hlOSM on infarcted hearts of mice. Repetitive, intraperitoneal injections after the onset of MI as well as on Days 1, 2 and 3 post-MI were performed to test and compare the cardioprotective potency of an externally supported activation of LIFR and OSMR-mediated signaling in the post-ischemic murine heart (Figure 5A). A dose of 100 ng recombinant mLIF, mOSM and hlOSM per gram of body weight and day was chosen as equivalent doses of murine OSM have been shown to improve cardiac performance and survival of infarcted WT mice in prophylactic pretreatment approaches [5]. We first characterized modulatory effects on STAT3, STAT5 and c–Myc activation within the non-infarcted remote zone (RZ), border zone (BZ) and infarction zone (IZ) at Day 4 post-MI to (i) prove a cardiac-specific effect by our systemic delivery approach and (ii) perform a detailed spatial expression analysis of signaling molecules activated in cultured cardiomyocytes upon treatment with our battery of cytokines (Figure 5A).

In comparison to the +PBS reference group, we could identify an increased phosphorylation of STAT3 within the RZ, BZ and IZ upon injection of mLIF, mOSM and hlOSM (Figure 5B,C). The administration of mLIF and hlOSM resulted in a higher activation profile of STAT3 when compared to mOSM and appeared equivalent with a more prominent increase of p–STAT3 (Tyr705) within the BZ and IZ (Figure 5B,C). The overall protein expression of STAT3 was not affected among all groups (Figure 5B, Appendix A). Very similar to phosphorylated STAT3, we found an increased activation of p–STAT5 via mLIF and hlOSM in infarcted hearts of mice (Figure 5D,E). Importantly, a significant increase of p–STAT5 (Tyr694) by mLIF and hlOSM was mostly referred to the non-infarcted remote zone, which is diametrically opposed to the increase of p–STAT3 (Tyr705) within infarcted and ischemically challenged regions (Figure 5D,E). Finally, the increased activation profile of STAT5 in RZ and BZ was accompanied with a moderate increase of total STAT5 protein expression by mLIF and hlOSM (Figure 5D, Appendix A). 

The most significant changes in total protein expression upon systemic injections of mLIF, mOSM and hlOSM were identified for c–Myc within the IZ at Day 4 post-MI (Figure 5F, Appendix A). Virtually all cytokines mediated a pronounced increase of total c–Myc protein levels, whereas its activation appeared to be very specific to murine OSM (Figure 5F,G and Appendix A). More detailed, systemic injections of mOSM but not mLIF and hlOSM resulted in a massive phosphorylation of c–Myc throughout the entire myocardium (Figure 5F,G). 

The differential expression of phosphorylated STAT3, STAT5 and c–Myc upon systemic injections of mLIF, mOSM and hlOSM substantiate the validity of this approach to modulate cardiac-specific processes. Unexpectedly, our spatial expression analysis of signaling molecules imply a complex and non-redundant activity of OSM and LIF at distinct sites of the injured heart.

### 2.5. Simultaneous Activation of the OSMR and LIFR by hlOSM after the Onset of MI Preserves Cardiac Architecture and Contractility

We subsequently studied potential morphological and functional consequences of our post-infarction cytokine administration approach in mice via magnetic resonance imaging (MRI) at Day 28 post-MI (Supplementary. Appendix A). Accordingly, we determined a lower end-diastolic and end-systolic volume in hlOSM– versus PBS–, mLIF– and mOSM–treated mice (Appendix A). The stroke volume was comparable between all groups (Appendix A). A reduced left ventricular mass, smaller infarct size and a higher left ventricular ejection fraction (LVEF) suggested a better preserved left ventricular architecture of mice receiving hlOSM, although these differences were not statistically significant (Appendix A). As these parameters, particularly LVEF, mostly reflect alterations in global ventricular dimensions and not actual myocardial contractility, we performed a more detailed characterization of the left ventricle by examining myocardial strain. This emerging technique allows the targeted study of the longitudinal, circumferential, and radial directed contractile myocardial deformation at both global and regional levels and has been shown to be more intimately related to cardiomyocyte contractility than LVEF (Figure 6A) [28,29,30]. Referring to our four groups, we found higher negative global longitudinal (GLS) and global circumferential strain (GCS) peak-systolic strain values in hlOSM-treated mice, indicating a more preserved contractile function of the left ventricle after MI (Figure 6B,C). Mean global radial strain (GRS) values, a parameter for wall thickening, were not different between PBS and cytokine-treated mice (Figure 6D). 

Given the structural and functional differences between infarcted and non-ischemic regions, we additionally performed regional circumferential strain (RCS) analysis of the left ventricle being divided into an infarcted apical part, adjacent midventricular part, and a remote basal part (Figure 6E). Here, we could identify a higher negative regional circumferential strain value not only in the injured apical and midventricular section but also in the non-injured basal regions of infarcted mice, which had received hlOSM (Figure 6F–I). Repetitive injections of mLIF and mOSM did not affect regional circumferential strain values, i.e., were not efficient in preserving cardiac contractility within defined segments when compared with PBS-treated mice (Figure 6F–I). We further correlated RCS values with respective infarct sizes to elaborate whether the improved contractile function of the remote basal myocardium is referred to a more preserved global architecture or to an improvement in local myocyte function (Appendix A). Pearson correlations and linear regression of RCS values with infarct size showed a strong positive correlation between regional apical circumferential strain values and the size of myocardial infarction (r = 0.70, *p* < 0.0001) (Appendix A). This relationship progressively weakened from the midventricular (r = 0.61, *p* = 0.0001) to the basal section (r = 0.41, *p* = 0.02) (Appendix A). This marked decrease in correlation of infarct size and RCS from the infarcted apical portion to the remote basal myocardium conclusively suggests that hlOSM exerts not only direct cardioprotective on hypoxic but on normoxic cardiomyocytes as well. A simultaneous and rapid activation of both OSMR and LIFR within infarcted hearts might therefore constitute a novel opportunity to preserve the functional and structural integrity of the damaged heart.

## 3. Discussion

Loss- and gain-of-function studies in animal models of ischemic heart disease revealed a crucial role of OSMR and LIFR activation for cardiac repair [6,7,8,9,10]. These promising basic findings, however, have not been translated into clinical trials and applications mostly because of species-specific receptor binding diversities of OSM in mice and humans. The prophylactic and long-term design of most delivery approaches to enhance OSM and LIF signaling in rodents additionally did not permit a reliable prognosis of their therapeutic potential for the treatment of acute ischemic heart diseases. Our present study especially aimed to bridge this gap via (i) comparing downstream effects of exclusive LIFR, OSMR and dual LIFR/OSMR signaling in murine cardiomyocytes via administration of recombinant murine LIF (mLIF), murine OSM (mOSM) and an engineered human-like OSM mutant protein (hlOSM), (ii) monitoring the endogenous kinetic expression profile of OSM and LIF to define a clinically relevant time frame for putative therapeutic interventions and (iii) assessing the cardioprotective potency of an externally supported activation of LIFR and OSMR-mediated signaling in the post-ischemic murine heart via post-infarction delivery of recombinant mLIF, mOSM and hlOSM.

Our data suggest that a simultaneous and rapid activation of both receptors throughout the inflammatory phase of myocardial healing effectively reduces cell death of cardiomyocytes within the infarcted myocardium. This direct cardioprotective effect of hlOSM administration appears to involve not only the myocardial tissue downstream of the occluded left anterior descending artery but encompasses midventricular and basal regions, as suggested by our resolution of regional myocardial strain and correlation analyses. In other words, repetitive short-term injections of recombinant hlOSM are sufficient to reduce infarct expansion in mice, which is originally defined as the combination of left ventricular wall thinning in the radial direction and dilation in the longitudinal–circumferential plane [31,32]. Unaltered global radial strain values of PBS– versus hlOSM-treated mice do not pinpoint to an altered diameter, i.e., thinning of the left ventricle. A hypertrophic effect of OSMR– and LIFR-mediated signaling on cardiomyocytes under acute ischemic conditions, as suggested previously, also seems unlikely in view of our data [33,34]. The preserved contraction in longitudinal and circumferential directions of hlOSM-treated mice rather suggest an increased cardiomyocyte resistance to hypoxia, which has been similarly observed in our hypoxic cardiomyocyte cultures upon stimulation with mLIF, mOSM and hlOSM. The overall importance of a restricted infarct expansion is not only related to protect contracting cardiomyocytes from cellular death but to diminish progressive remodeling culminating in an increased incidence of cardiac rupture, left ventricular dilation and heart failure [35,36,37].

The simultaneous but divergent activation profile of STAT3 within infarcted and non-infarcted regions on the one hand and activation of STAT5 restricted to non-infarcted regions on the other hand imply site-specific and environmentally dependent activities of mOSM, mLIF and hlOSM on cardiomyocytes. Normoxic and hypoxic cultures of murine cardiomyocytes substantiate this hypothesis as our battery of cytokines is able to activate STAT3 under both conditions, whereas STAT5 is found to be exclusively activated under normoxic conditions. The activation of different STATs by one cytokine has been documented in numerous cell types and has been shown to allow fine adjustments of cellular processes [38]. Cardiomyocyte-specific deletions of either STAT3 or STAT5 in mice subjected to models of myocardial ischemia/reperfusion emphasized their requirement for cardiomyocyte survival via upregulation of anti-apoptotic and anti-oxidant proteins [18,39,40,41]. Certainly, the improved adaptability of cultured cardiomyocytes to survive hours of ischemia upon treatment with mLIF, mOSM and hlOSM additionally argue for rapid, non-transcriptional processes orchestrating cellular survival. Indeed, ischemic conditions have been recently reported to induce intracellular translocation of STAT3 to mitochondria in cardiomyocytes, which inhibits the opening of the mitochondrial permeability transition pore and thus allows the cell to better cope with oxidative stress [42,43,44]. Mitochondrial activities of STAT5 have been reported as well, but its contribution to protect ischemically challenged cardiomyocytes has not been studied yet [45]. A reduced expression of hypoxia inducible factor−1α, interleukin−10 and vascular endothelial growth factor in hearts of cardiomyocyte-specific STAT5 knockout mice following remote ischemic preconditioning combined with myocardial ischemia/reperfusion injury imply a complementary and indirect cardioprotective regulatory role of STAT5 during cardiac repair [39]. The increase of active p–STAT5 in remote, non-infarcted regions upon mLIF and hlOSM administration may conclusively constitute a LIFR-mediated indirect protection loop, which adds to the STAT3–mediated resilience and survival of hypoxic cardiomyocytes. Distinct activation profiles of STAT3 and STAT5 in pig models of regional ischemia/reperfusion injury and remote ischemic preconditioning in human patients undergoing coronary artery interventions alternatively prompted an association of STAT3 activation and STAT5 inhibition with increased cardioprotection [22,46]. The simultaneous activation of STAT3 and STAT5 in normoxic cardiomyocytes by mLIF, mOSM and hlOSM, however, does not necessarily speak for such opposing or even antagonizing regulatory circuits.

Our transcriptome and immunoblot analysis of cultured cardiomyocytes and infarcted tissue characterized c–Myc as a third transcription factor being involved in LIFR- and OSMR-mediated downstream signaling. A cardioprotective function of c–Myc has been referred to its modulatory role on cardiac metabolism under ischemic conditions, which is marked by a switch from fatty acid oxidation to glycolysis [47]. Accordingly, activation of c–Myc in ischemic hearts was shown to increase glucose utilization and decrease fatty oxidation in order to meet the metabolic energy demand of cardiomyocytes during oxygen shortage [48]. This switch in cardiac metabolism is further supposed to contribute to the phenotypic conversion of cardiomyocytes towards a dedifferentiated state after the onset of myocardial ischemia [49]. The potency of OSM to induce cardiomyocyte dedifferentiation along with the predominant activation of c–Myc in mOSM–treated mice strongly suggest an intracellular interplay between glycolysis and dedifferentiation via OSM–OSMR signaling [5]. Conversely, our findings imply that the activation of LIFR–mediating signaling cascades within infarcted hearts do not result in a significant activation of c–Myc. The increased activation of STAT3 by mLIF and hlOSM putatively pose a negative regulatory module of c–Myc activation as STAT3 has been shown to suppress effector functions of c–Myc in fibroblasts [50].

An enhanced but divergent activation profile of STAT3 and STAT5 via mLIF and hlOSM versus increased activation of c–Myc upon mOSM administration conclusively argue for ligand–receptor–specific activation of distinct signaling events within the injured heart. A preserved cardiac function following administration of hlOSM but not mLIF exemplifies the need to further dissect those molecular mechanisms, by which a combined activation of OSMR and LIFR protects the myocardium from ischemia-mediated cell death.

## 4. Materials and Methods

### 4.1. Recombinant Proteins

Expression and purification of recombinant native murine LIF (mLIF), native murine OSM (mOSM) and the chimeric human-like OSM mutant protein (hlOSM) were performed as described previously [17]. All recombinant proteins added to cardiomyocyte cultures or administrated via intraperitoneal injections into mice were diluted in sterile PBS. 

### 4.2. Cardiomyocytes Isolation and Cultivation

Isolation of adult murine cardiomyocytes was performed by using standard procedures [51]. Afterwards, cells were cultured under normoxic (20% O_2_, 5% CO_2_, 94%N) and hypoxic conditions (1% O_2_, 5% CO_2_, 94%N) by using a conventional CO_2_ Incubator (Heracell 150) and hypoxic chamber (HypOxystation) for 24 h, respectively. Stimulation of cardiomyocytes with recombinant mLIF, mOSM and hlOSM (each at a final concentration of 20 ng mL^−^^1^) were carried out in serum-free medium supplemented with 1% Pen/Strep. 

### 4.3. Cell Viability Assays

Cells were stimulated with 20 ng mL^−^^1^ of the indicated cytokines for 12 h, then exposed to normoxic (20% O_2_, 5% CO_2_, 94%N) or hypoxic conditions (1% O_2_, 5% CO_2_, 94%N) for an additional 12 h, after which cells were stained with Calcein-AM (Invitrogen) following the manufacturer’s instructions. Briefly, wells were washed once with HBSS before adding a 2 µM Calcein-AM solution and Hoechst 33342 Solution (20 mM) in HBSS and incubated for 30 min at 37 °C. The total number of viable Calcein-AM positive cells were quantified by using ImageXpress Micro XLS Widefield High-Content Analysis System (Molecular Devices).

The release of lactate dehydrogenase by hypoxic cardiomyocytes was analyzed by using the CytoTox 96^®^ Non-Radioactive Cytotoxicity Assay (Promega) and performed according to the manufacturer’s instructions.

### 4.4. Myocardial Infarction and Recombinant Protein Administration

Adult male C57BL/6J mice at the age of 10–12 weeks (25–30 g) were subjected to permanent ligation of the left artery descending (LAD) artery as described previously [52]. 

Repetitive intraperitoneal injections of recombinant mLIF, mOSM and hlOSM (100 ng per gram of body weight and day, diluted in 100 µL sterile PBS) were performed with a 33-gauge needle after the onset of MI as well as on Days 1, 2 and 3 post-MI. Injection of equivalent volumes of sterile PBS served as control. All animal experiments were performed in accordance with national German animal protection laws and EU (Directive 2010/63/EU) ethical guidelines and were approved by the local governmental animal protection committee Regierungspräsidium Darmstadt.

### 4.5. Cardiac Magnetic Resonance Imaging

All cardiac imaging experiments were carried out on 7.0 T Bruker Pharmascan (Bruker, Ettlingen, Germany), equipped with a 760 mT/m gradient system, using a cryogenically cooled four channel phased array element 1H receiver-coil (CryoProbe) and a 72 mm room temperature volume resonator for transmission and the IntraGateTM self-gating tool. Measurements are based on the gradient echo method with a repetition time = 6.2 ms; echo time = 1.3 ms; field of view = 2.20 × 2.20 cm; slice thickness = 1.0 mm; matrix = 128 × 128; number of frames = 14. Two-chamber long-axis view, four-chamber long-axis view, and six to seven short-axis planes to cover the left ventricle were acquired. Mice were measured under volatile isoflurane (1.5–2.0% in oxygen/air with a flow rate of 1.0 L/min) anesthesia; the body temperature is maintained 37 °C by a thermostatically regulated water flow system during the entire imaging protocol. Volumetric and functional analysis were performed using Medis Suite, QMass 3.2.60.4 (Medis Medical Imaging System, Leiden, The Netherlands). For cardiac strain analysis the feature tracking technology in the Medis Suite QStrain 3.2 module was used. Values for global longitudinal peak systolic strain (GLS) were determined at long-axis (LAX) slices of a 4-chamber view orientation and values for global (GCS) as well as for regional circumferential peak systolic strain (RCS) were determined at short-axis (SAX) slices. Global radial peak systolic strain (GRS) values were measured on LAX slices of a 4-chamber view orientation and on SAX slices and subsequently averaged. The size of the infarcted myocardium was assessed as described elsewhere [53].

### 4.6. RNA Sequence Analysis and Gene Set Enrichment Analysis

RNA was isolated from cultured cardiomyocytes via using the miRNeasy micro-Kit (Qiagen) combined with on-column DNase digestion (DNase-Free DNase Set, Qiagen) to avoid contamination by genomic DNA. RNA and library preparation integrity were verified with LabChip Gx Touch 24 (Perkin Elmer). Further, 1 µg of total RNA was used as input for SMARTer Stranded Total RNA Sample Prep Kit-HI Mammalian (Clontech). Sequencing was performed on the NextSeq500 instrument (Illumina) using v2 chemistry, resulting in average of 25 M reads per library with 1 × 75 bp single end setup. The resulting raw reads were assessed for quality, adapter content and duplication rates with FastQC (available online at: http://www.bioinformatics.babraham.ac.uk/projects/fastqc, accessed on 4 October 2021). 

Trimmomatoc version 0.39 was employed to trim reads after a quality drop below a mean of Q20 in a window of 10 nucleotides. Only reads between 30 and 150 nucleotides were cleared for further analyses. Trimmed and filtered reads were aligned versus the Ensembl mouse genome version mm10 (ensemble release 101) using STAR 2.7.7a with the parameter “--outFilterMismatchNoverLmax 0.1” to increase the maximum ratio of mismatches to mapped length to 10% [54]. The number of reads aligning to genes was counted with featureCounts 1.6.5 tool from the Subread package [55]. Only reads mapping at least partially inside exons were admitted and aggregated per gene. Reads overlapping multiple genes or aligning to multiple regions were excluded. The Ensemble annotation was enriched with UniProt data based on Ensembl gene identifiers (Activities at the Universal Protein Resource (UniProt).

Normalized RNAseq counts thus obtained were used as input to perform gene set enrichment analysis (GSEA) using GSEA v4.1.0 and the hallmark gene set collection obtained from the Molecular Signatures Database (MSigDB) [56,57]. Default analysis parameters were employed except for permutation type, which was changed to “gene set” as recommended for low numbers of biological replicates, and the resulting gene enrichment scores and adjusted *p*-values were plotted using the ggplot2 package in R (available online at https://ggplot.tidyverse.org and https://ggplot.tidyverse.org and http://www.R-project.org/, accessed on 4 October 2021).

### 4.7. Protein Extraction and Immunoblot Analysis

Proteins derived from cardiomyocyte cultures and myocardial tissues were isolated by using lysis buffer (0.1 M Tris-HCl pH 8.8, 0.01 M EDTA, 0.04 M DTT, 10% SDS, pH 8.0) supplemented with protease inhibitors (500 μg mL^−1^ Benzamidin, 2 μg mL^−1^ Aprotinin, 2 μg mL^−1^ Leupeptin, 2 mM PMSF, 1 mM Sodium Vanadate, 20 mM Sodium Fluoride). Proteins were separated by SDS-PAGE on Gradient NuPAGE 4–12% Bis-Tris gels (Invitrogen) and blotted onto nitrocellulose membranes (Invitrogen). Subsequently, membranes were probed with the following specific primary antibodies: goat anti-mouse OSMR (R&D Systems; Catalogue No. AF-662), goat anti-human LIFR (R&D Systems, Catalogue No. AF-249), goat anti-mouse OSM (R&D Systems, Catalogue No. AF-495), goat anti-mouse LIF (R&D Systems, Catalogue No. AF-449), rabbit anti-Phospho-Stat3 (Tyr705) (Cell Signaling, Catalogue No. 9131), rabbit anti-Stat3 (Cell Signaling, Catalogue No. 4904), rabbit anti-Phospho-Stat5 (Tyr694) (Cell Signaling, Catalogue No. 9359), rabbit anti-Stat5 (Cell Signaling, Catalogue No. 94205), rabbit anti-Phospho-c-Myc (Ser62) (Cell Signaling, Catalogue No. 13748) and rabbit anti c-Myc (Cell Signaling, Catalogue No. 5605). Secondary antibodies conjugated with horseradish peroxidase were purchased from R&D Systems. Immunoreactive proteins were visualized by chemiluminescence using SuperSignal™ West Femto Maximum Sensitivity Substrate (ThermoFisher, Waltham, MA, USA, Catalogue No. 34095) and a ChemiDoc™ MP Imaging System (Bio-Rad, Hercules, CA, USA). Image Lab software 5.0 (Bio-Rad) was employed for densitometric quantification for band intensities. Relative expression ratios (FC) between PBS- and cytokine-treated cardiomyocytes were additionally calculated, whereas protein expression in PBS-treated cardiomyocytes was set to 1.

### 4.8. Immunofluorescence

Hearts were harvested 7 days after MI, cryopreserved with 15% and 30% sucrose overnight at 4 °C, embedded in Tissue-Tek O.C.T (Sakura Finetek, catalogue no. 4583) and frozen at −80 °C. Slides containing 20 µm transversal cryosections of the infarcted apex were frozen at −20 °C. Histological slices were permeabilized with 0.1% Triton X-100 (Sigma) for 10min at room temperature and blocked with PBS 3% BSA for 1 h. For immunostainings, the following primary antibodies were diluted in PBS 1% BSA and probed overnight at 4 °C: Goat anti-OSMR (RnD Systems, Catalogue No. AF662; 4 µg/mL) and Rabbit anti-LIFR/CD118 (Bioss, catalogue no. bs-1458R; 10 µg/mL). After extensive washing with PBS, secondary antibodies Donkey anti-Goat 647 (Invitrogen, Catalogue No. A21447 1:300) and Donkey anti-Rabbit 647 (Invitrogen, Catalogue No. A31573 1:300) were respectively added for 1 h at room temperature. Sections containing secondary but not primary antibodies were used as negative controls (not shown). The preserved myocardium was counterstained with Phalloidin-FITC (Sigma, catalogue no. P5282 1:200) for 1 h at RT and slides were mounted with Mowiol (Sigma). Confocal images were acquired using a SP8 Confocal Laser Scanning microscope (Leica) and maximum intensity projections derived using the Fiji software.

### 4.9. Statistical Analysis and Structural Visualization

Statistical analysis was performed with GraphPad Prism (GraphPad Software, San Diego, CA, USA). One-way ANOVA for one and two-way ANOVA for two independent variables with subsequent Bonferroni multiple comparisons were used for comparison of three or more groups. *p*-values less than 0.05 were considered statistically significant (*p* < 0.05). Homology models of mOSM, mLIF and hlOSM were constructed with SWISS-MODEL (10.1093/nar/gku340) and visualized with MacPymol (version 1.7.2.1, Schrödinger LLC, New York, NY, USA).

## Figures and Tables

**Figure 1 ijms-23-00353-f001:**
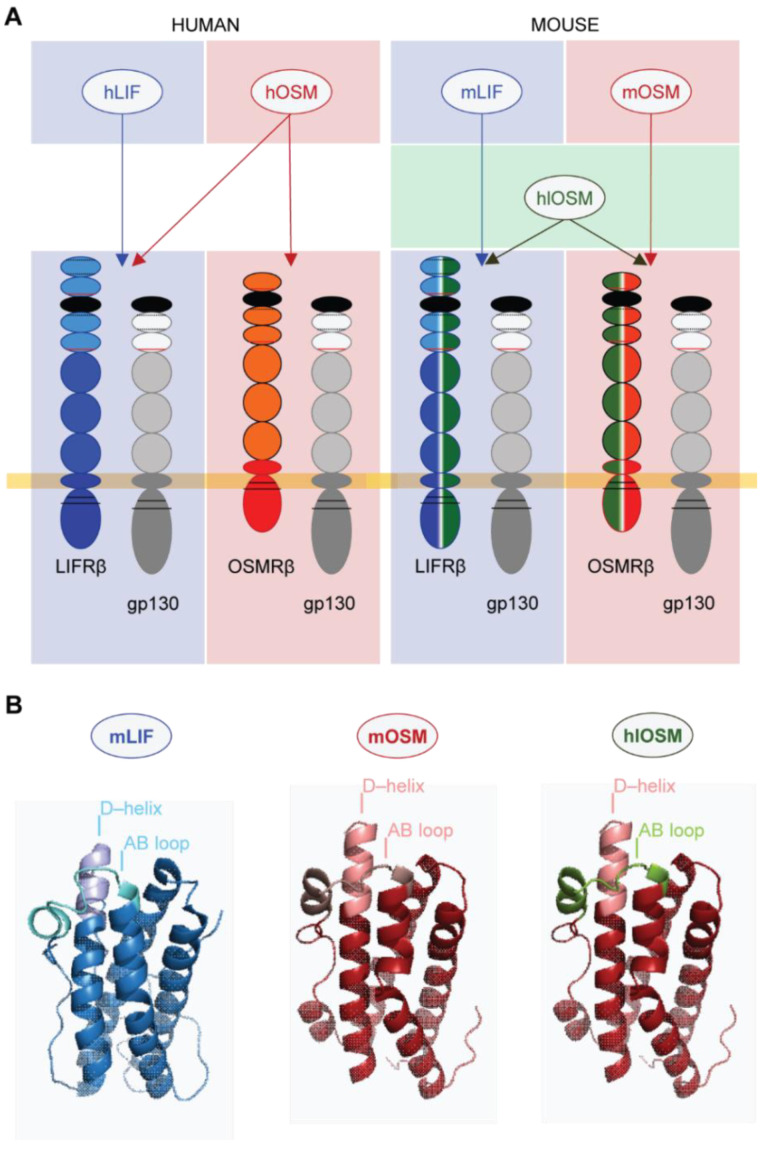
Species-specific binding differences of LIF and OSM to OSMR and LIFR in human and mice. (**A**) Schematic illustration of species-specific binding properties of human OSM (hOSM), human LIF (hLIF), murine OSM (mOSM) and murine LIF (mLIF). Note that the generation of a human-like OSM (hlOSM) mutant mimics the binding properties of hOSM with the OSMR and LIFR in mice [17]. (**B**) Three-dimensional model of mOSM, mLIF and hlOSM. The AB loop and D–helix of each molecule constitute structural determinants of their species-specific receptor binding properties. The hlOSM protein contains the human AB loop sequence, which is highlighted in green.

**Figure 2 ijms-23-00353-f002:**
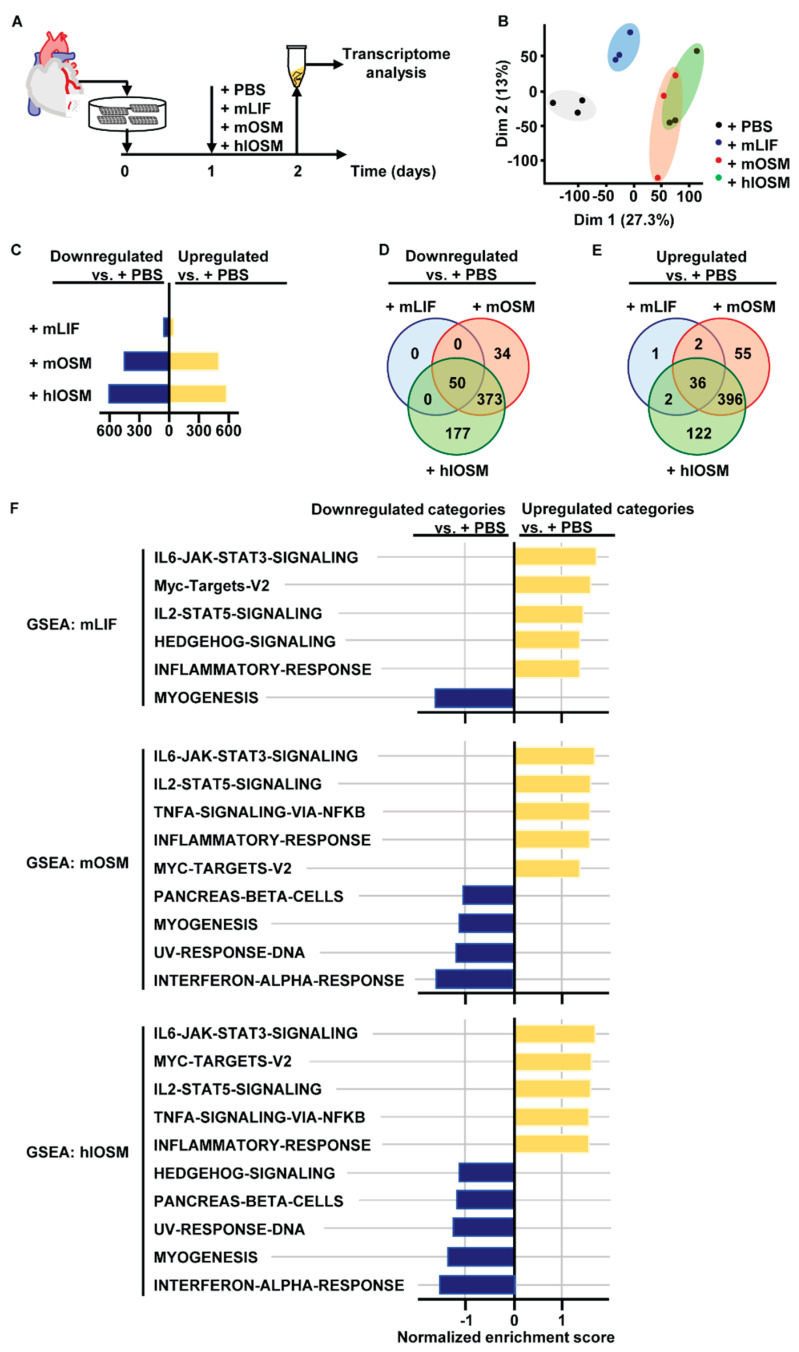
Transcriptome and gene set enrichment analysis characterizes STAT3, STAT5 and c–Myc as major common signaling molecules downstream of OSMR and LIFR activation in cardiomyocytes. (**A**) Experimental set-up for transcriptome analysis of primary murine cardiomyocytes after treatment with mLIF, mOSM and hlOSM (each at 20 ng mL^−1^) for 24 h (*n* = 3). Addition of equivalent volumes of sterile PBS served as control. (**B**) Principal component analysis of cultured cardiomyocytes treated with PBS, mLIF, mOSM and hlOSM. (**C**) Number of differentially up- and downregulated genes in mLIF–, mOSM– and hlOSM–treated versus PBS-treated cardiomyocytes. (**D**,**E**) Venn diagram of down- and up-regulated differentially expressed genes (DEG) in mLIF–, mOSM– and hlOSM–treated cardiomyocytes. (**F**) Gene set enrichment analysis (GSEA) of mLIF–, mOSM– and hlOSM–treated cardiomyocytes. GSEA was performed by pairwise comparisons of all cytokine-treated versus control samples. Gene sets are ranked by a normalized enrichment score.

**Figure 3 ijms-23-00353-f003:**
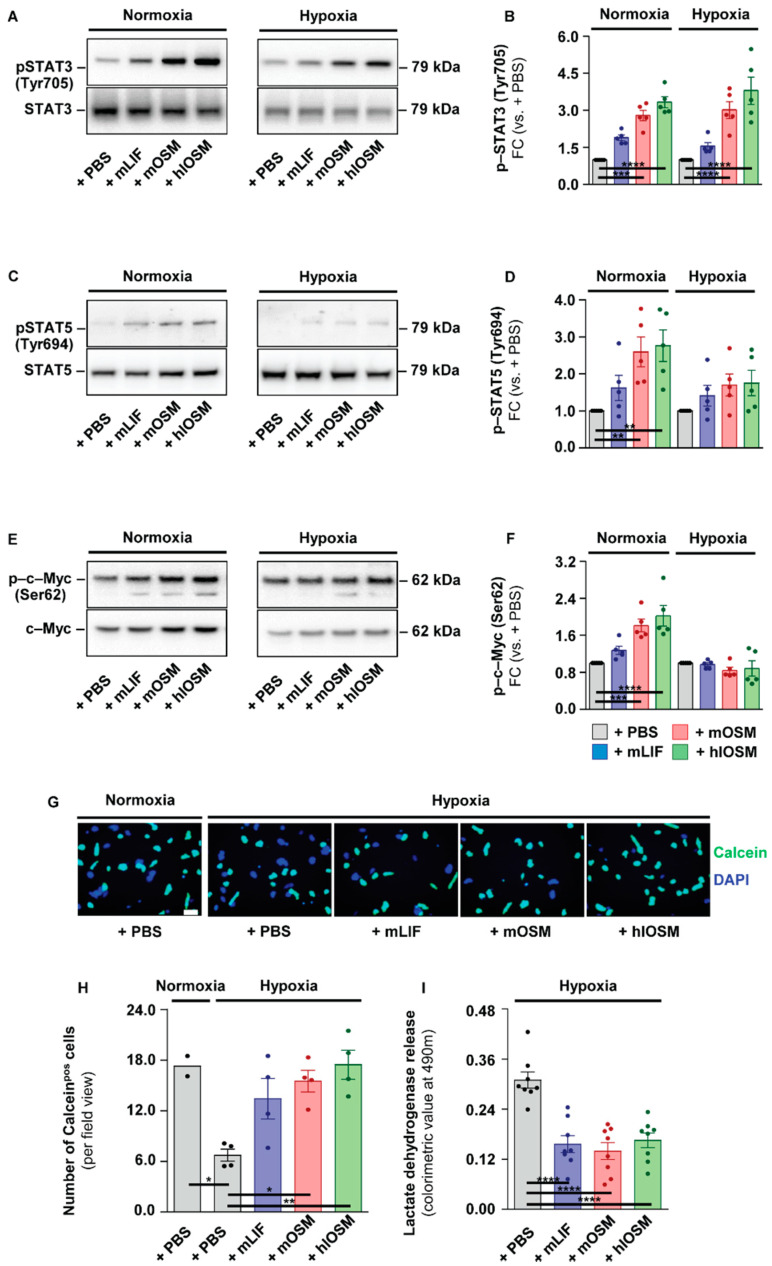
OSMR- and LIFR-mediated activation of STAT3 but not STAT5 and c–Myc coincides with an increased survival of cultured cardiomyocytes under hypoxic conditions. Representative images and semi-quantitative analysis of immunoblots of (**A**,**B**) phosphorylated (p–) STAT3 (Tyr705), (**C**,**D**) p–STAT5 (Tyr694) and (**E**,**F**) p–c–Myc (Ser62) in cardiomyocytes cultured under normoxic and hypoxic conditions (*n* = 5). Statistical analysis was performed by one-way ANOVA together with Bonferroni post-test comparisons to assess differences under normoxic and hypoxic conditions. Asterisks indicate Bonferroni post–hoc test significances between cytokine-treated versus PBS-treated cardiomyocytes with ** *p* < 0.01, *** *p* < 0.001 and **** *p* < 0.0001. Data represent ratios (FC) of expression in cytokine-treated versus PBS-treated cardiomyocytes. Total STAT3, STAT5 and c–Myc expression is shown below. (**G**,**H**) Representative immunofluorescent images and quantitative analysis of Calcein^pos^ viable cardiomyocytes (green) cultured under normoxic and hypoxic conditions (Normoxia +PBS, *n* = 2; Hypoxia +PBS, +mLIF, +mOSM and +hlOSM, *n* = 4). Hoechst 33342 (blue) visualizes nuclei of cardiomyocytes. (**I**) Statistical analysis of lactate dehydrogenase release by hypoxic cardiomyocytes was performed by one-way ANOVA (*n* = 8). Asterisks indicate Bonferroni post-hoc test significances between cytokine-treated versus PBS-treated cardiomyocytes as well as between PBS-treated cardiomyocytes under normoxic and hypoxic conditions with * *p* < 0.05, ** *p* < 0.01, *** *p* < 0.001 and **** *p* < 0.0001. Data are presented as mean ± sem.

**Figure 4 ijms-23-00353-f004:**
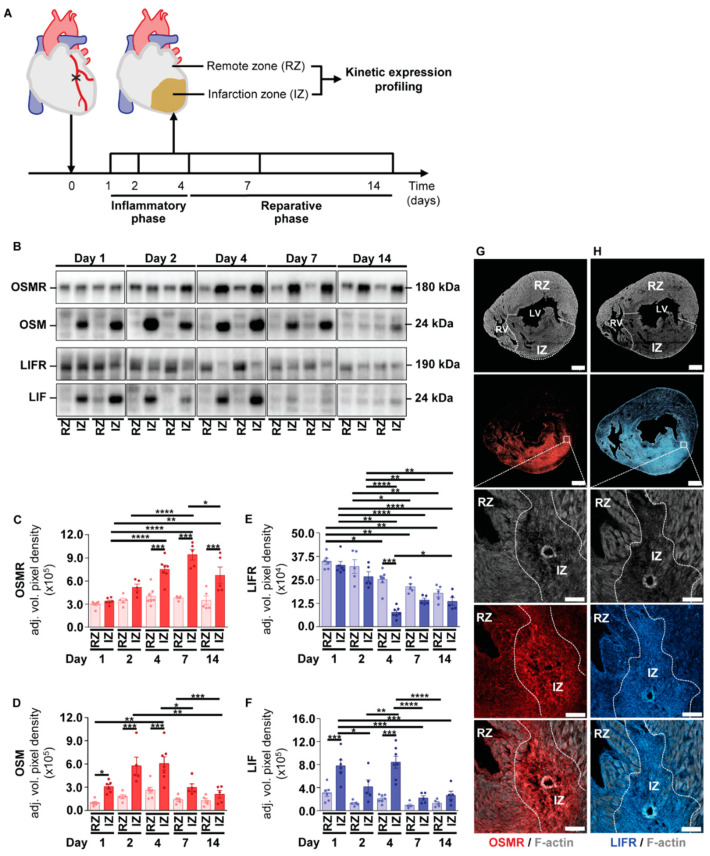
Kinetic expression pattern of OSM, LIF and their corresponding receptors OSMR and LIFR in cardiac tissue after the onset of myocardial infarction in mice. (**A**) Schematic illustration of kinetic expression profiling. Cardiac tissue was harvested at indicated time points post-MI and fractionated into a non-infarcted remote zone (RZ) and infarction zone (IZ). Expression of OSMR, OSM, LIFR and LIF in RZ and IZ was analyzed by immunoblotting (*n* = 6 at Day 1; *n* = 5 at Day 2; *n* = 7 at Day 4; *n* = 5 at Day 7; *n* = 5 at Day 14 post-MI). (**B**) Two representative immunoblots of OSMR, OSM, LIFR and LIF in RZ and IZ at indicated time points post-MI. (**C**–**F**) Semi-quantitative analysis of immunoblots shown in (**B**) based on the adjusted mean volume pixel density of bands. Statistical analysis was performed by two-way ANOVA. Asterisks indicate Bonferroni post-hoc test significances between RZ and IZ at individual time points as well as between RZ and IZ throughout the period of observation with * *p* < 0.05, ** *p* < 0.01, *** *p* < 0.001 and **** *p* < 0.0001. Data are presented as mean ± sem. Immunofluorescence analysis of (**G**) OSMR and (**H**) LIFR in infarcted hearts of mice 7 days after MI. F-actin: grey. OSMR: red. LIFR: blue. IZ: infarction zone. RZ: remote zone. RV: right ventricle. LV: left ventricle. Scale bars, 500 µm and 100 µm in magnified sections.

**Figure 5 ijms-23-00353-f005:**
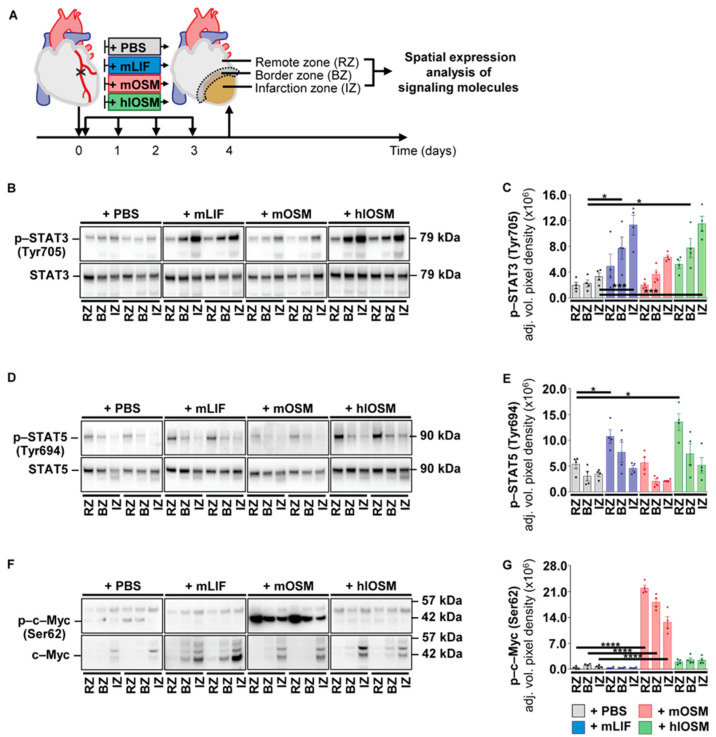
Post-infarction administration of mLIF, mOSM and hlOSM modulates the activation of STAT3, STAT5 and c–Myc at distinct sites of the myocardium. (**A**) Schematic illustration of systemic post-infarction administration of mLIF, mOSM and hlOSM in mice during the first three days of cardiac remodeling. At Day 4 post-MI, hearts were harvested, fractionated into a non-infarcted remote zone (RZ), border zone (BZ) and an infarction zone (IZ) to perform spatial expression analysis of signaling molecules via immunoblotting. Two representative images and semi-quantitative analysis of immunoblots of (**B**,**C**) p–STAT3 (Tyr705), (**D**,**E**) p–STAT5 (Tyr694) and (**F**,**G**) p–c–Myc (Ser62) in RZ, BZ and IZ of mice (*n* = 4 for all groups). Semi-quantitative analysis of immunoblots is based on the adjusted mean volume pixel density of bands. Statistical analysis was performed by one-way ANOVA together with Bonferroni post-test comparisons in order to monitor myocardial site-specific effects upon administration of PBS, mOSM, mLIF and hlOSM. Asterisks indicate Bonferroni post-hoc test significances between cytokine- and PBS-treated mice in RZ, BZ and IZ with * *p* < 0.05, *** *p* < 0.001 and **** *p* < 0.0001. Data are presented as mean ± sem.

**Figure 6 ijms-23-00353-f006:**
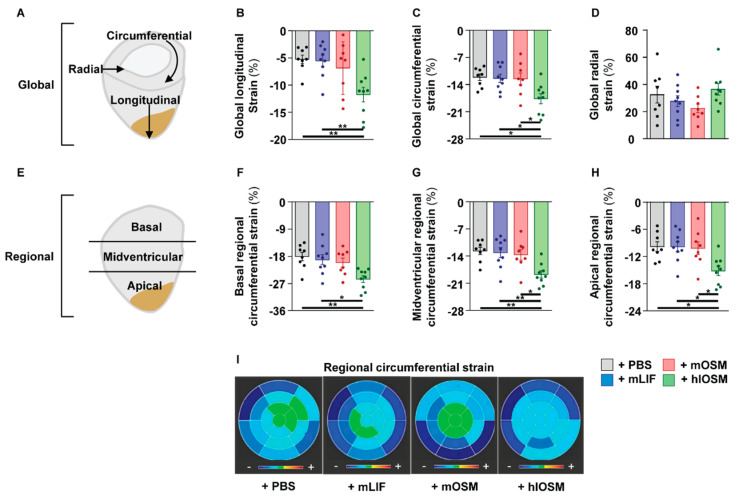
Simultaneous activation of the OSMR and LIFR by hlOSM preserves cardiac architecture and contractility after the onset of myocardial infarction. (**A**) Schematic illustration of the different global myocardial strain directions of the left ventricle: longitudinal shortening, circumferential shortening and radial thickening. Statistical analysis of magnetic resonance imaging-based (**B**) global longitudinal strain, (**C**) global circumferential strain and (**D**) global radial strain of the left ventricle at Day 28 post-MI following systemic injections of 100 ng recombinant mLIF, mOSM and hlOSM per gram of body weight at Days 0, 1, 2 and 3. Injections of equivalent volumes of sterile PBS, i.e., 100 μL per mouse, served as controls (+PBS, *n* = 8; +mLIF, *n* = 9, +mOSM, *n* = 8 and +hlOSM, *n* = 9). (**E**) Schematic illustration of the subdivision of the left ventricle into basal, midventricular, and apical segments used for regional circumferential strain (RCS) analyses. Statistical analysis of (**F**) basal RCS, (**G**) midventricular RCS and (**H**) apical RCS. Statistical analysis was performed by one-way ANOVA. Asterisks indicate Bonferroni post-hoc test significances between cytokine- and PBS–treated mice with * *p* < 0.05 and ** *p* < 0.01. Data are presented as mean ± sem. (**I**) Representative bull’s-eye plots illustrating the distribution of left ventricular regional circumferential strain values in PBS–, mLIF–, mOSM– and hlOSM–treated mice. The outer circle corresponds to basal, the middle circle to midventricular, and the inner circle to apical areas. Blue colors display negative strain values. Green colored areas reflect positive strain values.

## Data Availability

The data presented in this study are available in the article and supplementary material. Transcriptome data are available online at NCBI Gene expression omnibus (GSE185305, https://www.ncbi.nlm.nih.gov/geo/query/acc.cgi?acc=GSE185305, accessed on 4 October 2021).

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
