# Peer review of "Concomitant Activation of OSM and LIF Receptor by a Dual-Specific hlOSM Variant Confers Cardioprotection after Myocardial Infarction in Mice"

_ijms, 2021, doi:10.3390/ijms23010353_

Round 1

Reviewer 1 Report

This study showed the downstream effects of LIFR/OSMR signaling in cardiomyocytes and assessed the cardioprotective potency of LIFR and OSMR in MI model. The study design is clear and very interesting.

The expression of LIFR decreases at IZ in FIG. 4B, while it increases at IZ in FIG. 4H. Could you provide an enlarged view of the immunostaining of Figure 4G and H? It is useful in which cell expressed.

Only two dots can be confirmed in the + PBS bar in FIG. 3H. Since SEM cannot be calculated with n = 2, you should be corrected.

Line 440, Please add the information of the mouse train from which the myocardium is derived. Also, ref. 51 uses rats, but is it the same adjustment method?

Author Response

This study showed the downstream effects of LIFR/OSMR signaling in cardiomyocytes and assessed the cardioprotective potency of LIFR and OSMR in MI model. The study design is clear and very interesting.

The expression of LIFR decreases at IZ in FIG. 4B, while it increases at IZ in FIG. 4H. Could you provide an enlarged view of the immunostaining of Figure 4G and H? It is useful in which cell expressed.

Response: We have now provided a more detailed immunohistochemical analysis of OSMR and LIFR expression within infarcted hearts (see Fig 4. G, H). As stated within the manuscript, it becomes apparent that interstitial cells, e.g. fibroblasts, within the infarcted region and cardiomyocytes surrounding the infarcted region are immunopositive for both receptors.

Only two dots can be confirmed in the + PBS bar in FIG. 3H. Since SEM cannot be calculated with n = 2, you should be corrected.

Response: Of course, the reviewer is right in that matter. We have corrected the statistical analysis accordingly (see Fig. 3H).

Line 440, Please add the information of the mouse strain from which the myocardium is derived. Also, ref. 51 uses rats, but is it the same adjustment method?

Response: We have used adult male C57/BL6J mice for the isolation and cultivation of cardiomyocytes. We apologize for any confusion but the reference [51] “O'Connell, T.D., M.C. Rodrigo, and P.C. Simpson, Isolation and culture of adult mouse cardiac myocytes. Methods Mol Biol, 2007. 357: p. 271-96“ is referring to adult mouse cariomyocytes as well.

Reviewer 2 Report

This is a very elegant study which extends upon the work of the authors and others. The study is well-justified where the authors report a gap/limitation in the literature, and this study is designed to address said gap.

The study is quite thorough, well-designed, and the manuscript very well-written. The temporospatial analysis is a highlight.

Indeed, I felt I needed to be more critical of the paper and find more to comment on, but that became being critical for the sake of being critical and did the study/manuscript a disservice. 

That then leaves me with one comment - the use of statistical analysis needs to be better described and/or justified (especially for temporospatial analysis).

For example, Figure 3B: were all 8 groups subjected to the one one-way ANOVA or was an ANOVA done on the normoxic groups and another on the hypoxic groups? Was comparison made between normoxic and hypoxic groups? Depending on what is actually being assessed, one might argue that this should be a multi-way ANOVA.

Similarly for the graphs in figures 4 &5. Were all groups assessed within a single one-way? How was time/zone/treatment taken into account?

Author Response

This is a very elegant study which extends upon the work of the authors and others. The study is well-justified where the authors report a gap/limitation in the literature, and this study is designed to address said gap.

The study is quite thorough, well-designed, and the manuscript very well-written. The temporospatial analysis is a highlight.

Indeed, I felt I needed to be more critical of the paper and find more to comment on, but that became being critical for the sake of being critical and did the study/manuscript a disservice.

That then leaves me with one comment - the use of statistical analysis needs to be better described and/or justified (especially for temporospatial analysis).

For example, Figure 3B: were all 8 groups subjected to the one one-way ANOVA or was an ANOVA done on the normoxic groups and another on the hypoxic groups? Was comparison made between normoxic and hypoxic groups? Depending on what is actually being assessed, one might argue that this should be a multi-way ANOVA.

Similarly for the graphs in figures 4 &5. Were all groups assessed within a single one-way? How was time/zone/treatment taken into account?

Response: We have followed the reviewer´s advice and carefully rechecked our statistical analysis throughout the manuscript. Referring to Figure 3B, we have not performed a one–way ANOVA for all 8 groups as the normoxic and hypoxic groups were obtained from different origins. Hence, we used two separate one–way ANOVA for comparing normoxic and hypoxic groups. Concerning Figure 4, we have now performed a two-way ANOVA to take the different time-points into account. In addition to assess daywise differences between RZ and IZ, we have included spatiotemporal differences of OSMR, OSM, LIFR and LIF expression within the RZ and IZ (see Fig. 4C-F). With regard to Fig. 5, we have restricted our statistical analysis to spatial differences within the RZ, BZ and IZ in order to monitor myocardial site-specific effects upon administration of PBS, mOSM, mLIF and hlOSM. To this end, we have reasoned the choice of statistical analysis in detail within the corresponding figure legends of Fig. 3, Fig. 4 and Fig. 5.